# TAP: The Attention Patch for Cross-Modal Knowledge Transfer from Unlabeled Modality

**Yinsong Wang**                                                    *wang.yinso@northeastern.edu*
*Department of Mechanical and Industrial Engineering*
*Northeastern University*

**Shahin Shahrampour**                                          *s.shahrampour@northeastern.edu*
*Department of Mechanical and Industrial Engineering*
*Northeastern University*

**Reviewed on OpenReview:** *https://openreview.net/pdf?id=73uyerai53*

## Abstract

This paper addresses a cross-modal learning framework, where the objective is to enhance the performance of supervised learning in the primary modality using an unlabeled, unpaired secondary modality. Taking a probabilistic approach for missing information estimation, we show that the extra information contained in the secondary modality can be estimated via Nadaraya-Watson (NW) kernel regression, which can further be expressed as a kernelized cross-attention module (under linear transformation). This expression lays the foundation for introducing The Attention Patch (TAP), a simple neural network add-on that can be trained to allow data-level knowledge transfer from the unlabeled modality. We provide extensive numerical simulations using real-world datasets to show that TAP can provide statistically significant improvement in generalization across different domains and different neural network architectures, making use of seemingly unusable unlabeled cross-modal data.

## 1 Introduction

Consider a cross-modal learning framework where there exist a labeled primary modality and an unlabeled, unpaired secondary modality. In this paper, we address the following research question: can we boost the performance of supervised learning in the primary modality by exploiting the extra information in the secondary modality (that is unlabeled and unpaired with the primary modality)? Our work naturally lies at the intersection of cross-modal learning and semi-supervised learning. The cross-modal learning paradigm learns from data in different modalities (Li et al., 2018). From a probability theory perspective, cross-modal data often refers to data with different support dimensions and distribution curvatures. The semi-supervised learning paradigm uses unlabeled data to improve the model performance learned by limited labeled data (Zhu, 2005; Van Engelen & Hoos, 2020). While cross-modal learning and semi-supervised learning have been extensively studied independently, the intersection of the two has been less explored in the literature and remained elusive. To be specific, when we have a limited amount of labeled data in the primary modality and a large set of unlabeled and unpaired data in the secondary modality available during training, there is no principled learning paradigm that can make use of the secondary modality to create a model that is better than the model learned with only the primary modality. Figure 1a presents a visualization of our target problem to solve.

Our target problem is frequently encountered in different research communities. For example, when building a battery failure prediction model using temporal current and voltage readings, one option is to learn a supervised learning model with only current and voltage readings. However, there exist unlabeled videos of battery inner structure changes that are generated by different research labs, so a potentially more effective option is to incorporate the information of these videos into the model. These videos are information-rich,

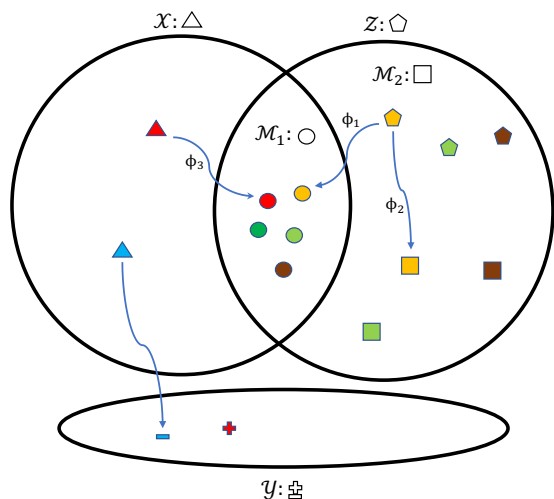
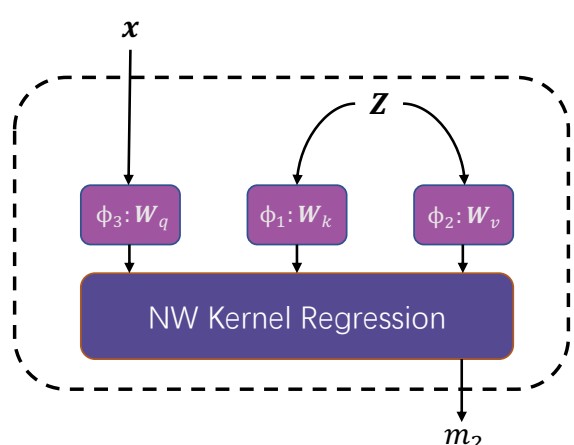

(a) For primary modality $\mathcal{X} \subseteq \mathbb{R}^{d_x}$ and the secondary modality $\mathcal{Z} \subseteq \mathbb{R}^{d_z}$, there exists a space $\mathcal{M}_1$ that contains the mutual information between $\mathcal{X}$ and $\mathcal{Z}$, where $\phi_1 : \mathcal{Z} \to \mathcal{M}_1$ and $\phi_3 : \mathcal{X} \to \mathcal{M}_1$ transform data $\mathbf{z} \in \mathcal{Z}$ and $\mathbf{x} \in \mathcal{X}$ to space $\mathcal{M}_1$, respectively. There also exists a space $\mathcal{M}_2$ that contains the exclusive information present in $\mathcal{Z}$ but not in $\mathcal{X}$. A transformation $\phi_2 : \mathcal{Z} \to \mathcal{M}_2$ takes data $\mathbf{z} \in \mathcal{Z}$ to $\mathbf{m}_2 \in \mathcal{M}_2$. The label information for data in $\mathcal{X}$ is available, but data in $\mathcal{Z}$ are unlabeled. There is also no alignment between data in $\mathcal{X}$ and $\mathcal{Z}$.

(b) The Attention Patch (TAP) parameterized by query transformation $\phi_3 : \mathbf{W}_q$, key transformation $\phi_1 : \mathbf{W}_k$, and value transformation $\phi_2 : \mathbf{W}_v$. TAP takes one instance $\mathbf{x} \in \mathcal{X}$ (or a representation of $\mathbf{x}$), uses a batch of reference data $\mathbf{Z} : \{\mathbf{z}_i \in \mathcal{Z}\}_{i=1}^{n_z}$ to generate the corresponding representation $\mathbf{m}_2 \in \mathcal{M}_2$, which contains information that is not present in space $\mathcal{X}$. The representation $\mathbf{m}_2$ will be concatenated with input $\mathbf{x}$ for downstream tasks.

Figure 1

but labeling them requires repeating the collection process, which is expensive and potentially impossible due to equipment constraints (Davidson et al., 2018). In general, reusing information-rich but hard-to-label datasets for different learning tasks calls for developing novel methods at the intersection of cross-modal learning and semi-supervised learning.

To solve the target problem, we start by formulating the missing information estimation problem from a probability density estimation perspective. Using kernel density estimation (Rosenblatt, 1956; Wand & Jones, 1994; Wang et al., 2023), we show that this formulation leads to a multivariate NW kernel regression, which can further be expressed as a kernelized cross-attention module (under linear transformation). We name this module The Attention Patch (TAP), as shown in Figure 1b. TAP is a simple neural network plugin that can be attached between two consecutive layers in a neural network. The TAP integration requires minimal modification to the original network (i.e., only moderate dimension change), and the parameters of mappings $\phi_1, \phi_2, \phi_3$ can be learned in parallel with the original network training.

## 1.1 Summary of Contributions

- This is the first work that investigates a learning paradigm in semi-supervised learning and cross-modal learning, where the "extra" information in $\mathcal{Z}$ is not only unlabeled and unpaired but also comes from a different modality than the primary modality $\mathcal{X}$. We propose a framework to enhance supervised learning of the primary modality $\mathcal{X}$ using unpaired, unlabeled secondary modality $\mathcal{Z}$.

- In Sections 3 and 4, we show that formulating our target problem as a missing information estimation problem leads to a multivariate Nadaraya-Watson (NW) kernel regression, and it further recovers a kernelized version of the popular *cross-attention* mechanism (Vaswani et al., 2017) (under linear transformation). Based on our observation, we propose The Attention Patch (TAP) neural network plugin for cross-modal knowledge transfer from unlabeled modality. We further propose a batch

training strategy to incorporate more unlabeled cross-modal data while maintaining moderate memory costs.

- We provide detailed simulations on three real-world datasets in different domains to examine various aspects of TAP and demonstrate that the integration of TAP into a neural network can provide statistically significant improvement in generalization using the unlabeled modality. We also provide detailed ablation studies to investigate the best configuration for TAP in practice, including the choice of kernel, the choice for latent space transformation, and compatibility with CNN and Transformer-based backbone feature extractors with an additional text-image dataset.

## 2 Related Literature

In this section, we provide a review of related literature that explains why existing techniques can not be directly applied to our target problem.

### 2.1 Cross-Modal Learning

Cross-modal learning focuses on learning with data from different modalities. The most representative topic application-wise is cross-modal retrieval. This topic focuses on finding relevant samples in one modality given a query in another modality (Wang et al., 2016). The most important step in cross-modal retrieval tasks is learning a coupled space that can correctly describe the correlation between data points from different modalities. Traditional techniques including canonical correlation analysis (Hardoon et al., 2004; Andrew et al., 2013; Wang & Shahrampour, 2021), partial least squares (Geladi & Kowalski, 1986; Cha, 1994), and bilinear model (Sharma et al., 2012; Tenenbaum & Freeman, 2000) find a simple projection of the matching pairs that minimize certain pre-defined loss functions. This framework, though has seen several variants and extensions (Ngiam et al., 2011; Hong et al., 2015; Sohn et al., 2014; Xu et al., 2015), remains the most popular framework in cross-modal correlation learning.

In addition to cross-modal retrieval, cross-modal supervised learning also follows the same principle. Examples include but are not limited to Lin & Tang (2006); Evangelopoulos et al. (2013); Jing et al. (2014); Feichtenhofer et al. (2016); Peng et al. (2017); Li et al. (2019). All of these models involve coupled space learning through either implicit or explicit learning loss with matching pairs of cross-modal inputs.

With the rise in popularity of natural language processing and time series analysis, a new concept of weak alignment appears in cross-modal learning. Weak alignment refers to the missing alignment of sub-components in instances from different modalities (Baltrušaitis et al., 2018). More specifically, this often means cross-modal input sequences with different sampling intervals or orders. For example, in the case of vision and language models, a text description sequence of a video will usually differ from the time and order of information that appeared in the video (Venugopalan et al., 2015; Tsai et al., 2019), or a text description of an image will need to map a sequence of words to a collection of objects without any specific orders (Mitchell et al., 2012; Kulkarni et al., 2013; Chen et al., 2015; Karpathy & Fei-Fei, 2015).

There exists a research direction that taps into instance-wise nonalignment for cross-modal learning, which is called non-parallel co-learning (Baltrušaitis et al., 2018). Non-parallel co-learning aims at improving the model learned on a single modality using another modality that is unaligned with the primary data. However, this is a concept that has only been studied with very specific applications, and it also requires the reference modality to be labeled during the training process. For example, cross-modal transfer learning (Frome et al., 2013; Kiela & Bottou, 2014; Mahasseni & Todorovic, 2016) mainly focuses on transferring supervised pre-trained embedding networks for improved cross-model prediction accuracy. Cross-modal meta-learning (Phoo & Hariharan, 2020; Islam et al., 2021) investigates improving few-shot learning performance of primary modality using a labeled, yet unaligned secondary modality.

*In a nutshell, almost all cross-modal learning frameworks focus on the case where there are known alignments between different modalities of data at least during the learning phase. Our work considers the case where both alignment and labels do not exist during learning, and our proposed architecture can be applied to arbitrary modalities.*

## 2.2 Semi-Supervised Learning

Semi-supervised learning focuses on addressing the challenge of limited labeled data availability in building a learning algorithm (Zhu, 2005; Van Engelen & Hoos, 2020). Among all kinds of semi-supervised learning methods, self-training is the closest class of methods that relates to our study. Self-training refers to the class of methods that train a supervised learning algorithm using labeled and unlabeled data together (Triguero et al., 2015). This approach is usually done by assigning pseudo labels to the unlabeled data and jointly refining the supervised learning model and the pseudo labels by iterative training (Yarowsky, 1995; Rosenberg et al., 2005; Dópido et al., 2013; Wu et al., 2012; Tanha et al., 2017). For most traditional learners, this means re-training the learning algorithm many times as the pseudo labels are being updated. However, this pseudo-label training approach naturally works with incremental learning algorithms like neural networks, where the model is gradually learned through optimizing an objective function (Lee et al., 2013; Berthelot et al., 2019; Zoph et al., 2020; Xie et al., 2020; Sohn et al., 2020). Strictly speaking, all pseudo-labeling/self-training methods focus on assigning labels in the prediction space to the data points in the primary space that is the same as the labeled data.

There exists another line of research that studies semi-supervised learning in the context of multi-view or multi-modal learning. For example, multi-view co-training methods (Blum & Mitchell, 1998; Kiritchenko & Matwin, 2001; Wan, 2009; Du et al., 2010) propose to train different classifiers on different views of the same data. Multi-view co-training generally assumes the two views are independent of each other but can perform equally well in a single-view prediction task. The development of deep neural networks also introduced a lot of cross-modal semi-supervised learning works, which mainly focus on computer vision and natural language processing (Nie et al., 2017b;a; 2019; Jia et al., 2020). Again, all of these mentioned works rely on the availability of labels in all the modalities.

*As we can see, all existing semi-supervised learning frameworks consider the case where the unlabeled data resides in the same space or joint space as the labeled data. Our work considers the case where the unlabeled data resides in a completely different space than the labeled data.*

## 3 Estimating the Missing Information

In this section, using kernel density estimators, we show that the missing information estimation problem coincides with the Nadaraya-Watson (NW) kernel regression. We further establish that this particular multivariate estimation scheme yields an asymptotically vanishing error.

### 3.1 Cross-Modal NW Kernel Regression

For clarity, we focus on estimating the missing information of one data point $\mathbf{x}$, which can be obtained from a data point $\mathbf{z}$ in another mode. We denote the corresponding representation of $\mathbf{x}$ in $\mathcal{M}_1$ as $\mathbf{m}_1 = \phi_3(\mathbf{x})$. Consider the missing information for $\mathbf{x}$ in space $\mathcal{M}_2$ as $\mathbf{m}_2 = \phi_2(\mathbf{z})$, which needs to be estimated with the help of reference dataset $\mathbf{Z} : \{\mathbf{z}_i \in \mathcal{Z}\}_{i=1}^{n_z}$. Let us write the conditional expectation of the missing information $\mathbf{m}_2$ given the representation $\mathbf{m}_1$ as follows

$$\mathbb{E}(\mathbf{m}_2|\mathbf{m}_1) = \int \mathbf{m}_2 \frac{p_1(\mathbf{m}_2, \mathbf{m}_1)}{p_2(\mathbf{m}_1)} d\mathbf{m}_2, \tag{1}$$

where $p_1(\mathbf{m}_2, \mathbf{m}_1)$ is the joint density of $\mathbf{m}_2$ and $\mathbf{m}_1$, and $p_2(\mathbf{m}_1)$ is the marginal density of $\mathbf{m}_1$. Using reference data $\mathbf{Z} : \{\mathbf{z}_i\}_{i=1}^{n_z}$, we apply kernel density estimation (Rosenblatt, 1956) for both above densities in the following form

$$p_1(\mathbf{m}_2, \mathbf{m}_1) \approx \frac{1}{n_z} \sum_{i=1}^{n_z} k(\phi_3(\mathbf{x}), \phi_1(\mathbf{z}_i)) k_1(\phi_2(\mathbf{z}), \phi_2(\mathbf{z}_i)) \triangleq \hat{p}_1(\phi_2(\mathbf{z}), \phi_3(\mathbf{x}))$$

$$p_2(\mathbf{m}_1) \approx \frac{1}{n_z} \sum_{i=1}^{n_z} k(\phi_3(\mathbf{x}), \phi_1(\mathbf{z}_i)) \triangleq \hat{p}_2(\phi_3(\mathbf{x})), \tag{2}$$

for proper kernel functions $k$ and $k_1$, which results in the following proposition.

**Proposition 1.** *The missing information estimation formulation in Equation* (1) *can be approximated with kernel density estimators in Equation* (2). *When the kernel function* $k_1(\cdot, \boldsymbol{\mu})$ *in Equation* (2) *is a density function for a distribution with mean* $\boldsymbol{\mu}$, *the approximation leads to*

$$\mathbb{E}(\mathbf{m}_2|\mathbf{m}_1) \approx \sum_{i=1}^{n_z} \frac{k(\phi_3(\mathbf{x}), \phi_1(\mathbf{z}_i))\phi_2(\mathbf{z}_i)}{\sum_{j=1}^{n_z} k(\phi_3(\mathbf{x}), \phi_1(\mathbf{z}_j))}. \tag{3}$$

The proof can be found in the Appendix. The proposition implies that the conditional expectation of missing information (given representation) leads to a multivariate version of the well-known NW kernel regression estimator (Nadaraya, 1964; Watson, 1964) that employs both modalities of data with the help of kernel density estimators. In Section 4.1, we will see that the above formulation recovers the popular kernelized cross-attention when $\phi_1, \phi_2, \phi_3$ are linear mappings.

## 3.2 Estimation Error Guarantee

We now investigate the estimation error of Equation (3) under several assumptions, especially for its connection to the reference sample size and the choice of the kernel function. In particular, we consider the case that $\mathbf{x}$ and $\mathbf{z}$ replace $\mathbf{m}_1$ (i.e., $\phi_3(\mathbf{x}), \phi_1(\mathbf{z})$) and $\mathbf{m}_2$ (i.e., $\phi_2(\mathbf{z})$), respectively. The NW kernel regression formulation in Equation (3) now becomes

$$\hat{f}(\mathbf{x}) = \sum_{i=1}^{n} \frac{k_{\mathbf{H}}(\mathbf{x}, \mathbf{x}_i)\mathbf{z}_i}{\sum_{j=1}^{n} k_{\mathbf{H}}(\mathbf{x}, \mathbf{x}_j)}, \tag{4}$$

where $k_{\mathbf{H}}(\mathbf{x}, \mathbf{x}_i) = \frac{1}{|\mathbf{H}|} k(\frac{\mathbf{x}-\mathbf{x}_i}{|\mathbf{H}|})$ is a kernel that satisfies the following mild technical assumption and the subscript of $n_z$ is dropped for convenience.

**Assumption 1.** (Wand & Jones, 1994) The bandwidth matrix $\mathbf{H} = h_n \mathbf{I}$ of the kernel function $k_{\mathbf{H}}(\cdot)$ with $|\mathbf{H}| = h_n^d$ (the subscript $n$ shows the dependence of $h$ to the number of data points) has the following properties

$$\begin{aligned} \lim_{n\to\infty} h_n &= 0, \\ \lim_{n\to\infty} n h_n^d &= \infty, \end{aligned} \tag{5}$$

which implies that the bandwidth parameter $h_n$ decays slower than $n^{-1/d}$ and converges to 0. The standard shift-invariant kernel function $k(\cdot)$ is a bounded, symmetric probability density function with a zero first moment and a finite second moment. That is, the following properties hold

$$\begin{aligned} \int_{\mathbb{R}^d} k(\mathbf{x})d\mathbf{x} &= 1, \quad \int_{\mathbb{R}^d} \mathbf{x}k(\mathbf{x})d\mathbf{x} = \mathbf{0}, \\ \int_{\mathbb{R}^d} \mathbf{x}\mathbf{x}^\top k(\mathbf{x})d\mathbf{x} &= \mu_2(k)\mathbf{I}, \\ \int_{\mathbb{R}^d} k^2(\mathbf{x})d\mathbf{x} &= R(k), \end{aligned} \tag{6}$$

where $\mu_2(k) < \infty$ and $R(k) < \infty$ are constants decided by the choice of kernel $k$.

It is easy to verify that several popular shift-invariant kernels $k(\mathbf{x} - \mathbf{y})$ (e.g., Gaussian kernel) satisfy the above assumption with proper normalization. We also put mild assumptions on the density functions and true mapping.

**Assumption 2.** The true density function $p(\mathbf{x})$ is differentiable and the $\ell_2$-norm of its gradient is bounded. The underlying true function $f : \mathcal{X} \to \mathcal{Z}$, (i.e., $\mathcal{M}_1 \to \mathcal{M}_2$) has a bounded gradient and Hessian in the norm sense, and we have $\mathbf{z} = f(\mathbf{x}) + \boldsymbol{\epsilon}$, where $\boldsymbol{\epsilon}$ is an isotropic noise vector from $\mathcal{N}(\mathbf{0}, \sigma^2\mathbf{I})$.

**Theorem 1.** *Under Assumption 1 and Assumption 2, the NW kernel regression estimator* $\hat{f}(\mathbf{x})$ *in Equation* (4) *with an isotropic shift-invariant kernel of bandwidth* $\mathbf{H} = h_n\mathbf{I}$ *yields an estimation error* $\mathbf{e}(\mathbf{x}) \triangleq \hat{f}(\mathbf{x}) - f(\mathbf{x})$

*that asymptotically converges in distribution as*

$$\sqrt{nh_n^d}\left(\mathbf{e}(\mathbf{x}) - \frac{h_n^{2d}\mu_2(k)\boldsymbol{\Psi}(\mathbf{x})}{p(\mathbf{x})}\right) \xrightarrow{d} \mathcal{N}(\mathbf{0}, \frac{R(k)\sigma^2}{p(\mathbf{x})}\mathbf{I}), \tag{7}$$

*where $\mathbf{x} \in \mathbb{R}^d$, and the $i$-th entry of $\boldsymbol{\Psi}(\mathbf{x})$ is*

$$\boldsymbol{\Psi}^{(i)}(\mathbf{x}) = \frac{1}{2}p(\mathbf{x})\,Tr\left[\nabla^2 f^{(i)}(\mathbf{x})\right] + Tr\left[\nabla f^{(i)}(\mathbf{x})\nabla p(\mathbf{x})^\top\right], \tag{8}$$

*where $\nabla^2 f^{(i)}(\mathbf{x})$ and $\nabla f^{(i)}(\mathbf{x})$ are the Hessian and gradient of $i$-th entry of the true function with respect to $\mathbf{x}$, $\nabla p(\mathbf{x})$ is the gradient of the true density function, and $Tr[\cdot]$ is the trace operator.*

The proof is provided in the Appendix. The above theorem shows that with proper kernel function, the estimation error $\mathbf{e}(\mathbf{x}) \to \mathbf{0}$ as $n \to \infty$ under Assumption 1 and Assumption 2. This suggests that under an ideal latent space transformation, more unlabeled data helps drive the estimation error to zero. We also observe that the error term $\mathbf{e}(\mathbf{x})$ converges to a sequence that scales with the factor $\mu_2(k)$, which implies kernel functions with lower $\mu_2(k)$ might contribute to a smaller $\mathbf{e}(\mathbf{x})$ in the non-asymptotic regime.

## 4 The Attention Patch

In this section, we formally propose The Attention Patch (TAP) by showing that the NW kernel regression formulation in Equation (3) with linear latent space transformation results in the popular "cross-attention" module (Vaswani et al., 2017). We then propose a batch training strategy for scalability.

### 4.1 Cross-Attention Module

In Equation (3), we showed how to estimate the missing information using the reference cross-modal data $\mathbf{Z}$. However, implementing this mechanism requires learning the latent space transformations $\phi_1, \phi_2, \phi_3$. Fortunately, a neural network integration of such formulation will allow the latent space transformation to be learned in parallel with the main learning objective. To be more specific, we can define the latent space transformations $\phi_1, \phi_2, \phi_3$ with learnable linear weight matrices $\mathbf{W}_k, \mathbf{W}_v, \mathbf{W}_q$, and we can see that the RHS of Equation (3) now becomes the kernelized version of the "cross-attention" module (Vaswani et al., 2017). Furthermore, as shown in Figure 2, TAP can be inserted between any two consecutive layers in a deep neural network with minimal modification to the original network. The integration process is equivalent to applying a patch to an existing neural network, hence the name The Attention Patch (TAP).

Now, we formally propose The Attention Patch in the following corollary.

**Corollary 1.** *For an output $\mathbf{x}$ of a layer in a deep neural network and an unlabeled cross-modal reference dataset $\mathbf{Z}$, the TAP integration means calculating the following*

$$\widehat{\mathbf{m}}_2 \approx \sum_{i=1}^{n_z} \frac{k(\mathbf{W}_q\mathbf{x}, \mathbf{W}_k\mathbf{z}_i)\mathbf{W}_v\mathbf{z}_i}{\sum_{j=1}^{n_z} k(\mathbf{W}_q\mathbf{x}, \mathbf{W}_k\mathbf{z}_j)}, \tag{9}$$

*and concatenating $\mathbf{x}$ and $\widehat{\mathbf{m}}_2$ for downstream tasks. The parameters $\mathbf{W}_q, \mathbf{W}_k, \mathbf{W}_v$ are learned in parallel with the original neural network. $k(\cdot, \cdot)$ is a shift-invariant kernel of choice, and a Gaussian kernel is recommended.*

Note that Equation (9) allows for easy implementation of TAP by feeding the whole set of reference data $\mathbf{Z}$ as keys and values in a cross-attention module during the training process. However, the latent space transformation can go beyond linear, as we will discuss later in the simulation.

### 4.2 Batch Training

The attention formulation of cross-modal learning in Equation (9) requires setting the keys and values to be the set of unlabeled reference data points $\mathbf{Z}$. Theorem 1 suggests that using more reference data for the model

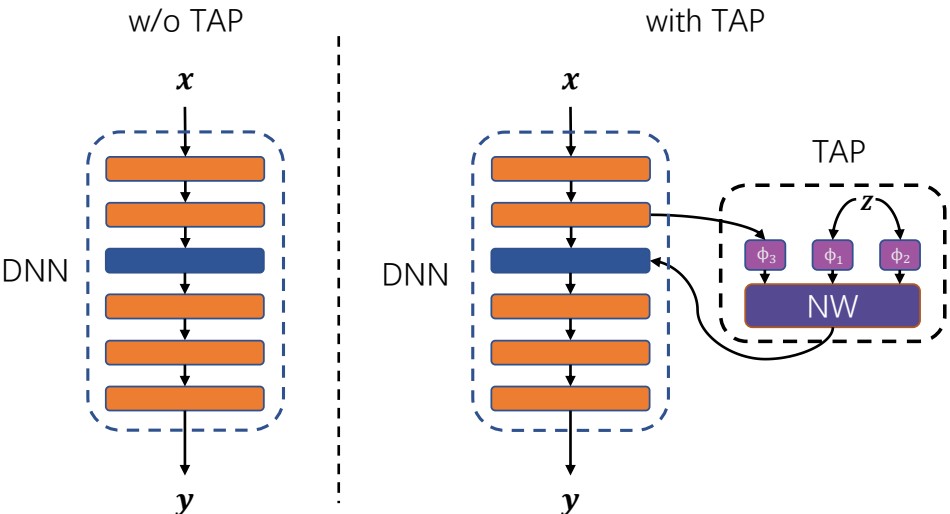

Figure 2: The Attention Patch (TAP) neural network integration visualization: TAP takes the output of a layer to calculate the missing representation using reference data $\mathbf{Z}$, and the output of TAP will be concatenated with TAP input and fed to the next layer. The only modification to the original deep neural network (DNN) is increasing the input dimension of the integration layer (blue layer).

will potentially result in a lower estimation error. However, the computation complexity of the cross-attention module scales linearly with respect to the sequence length of keys and values, which is equivalent to the number of reference points $n_z$ in $\mathbf{Z} \in \mathbb{R}^{n_z \times d_z}$. So, it is not practical to feed all the reference points at once, mainly due to memory limitations.

Therefore, we can break the reference dataset $\mathbf{Z}$ into $m$ batches $\{\mathbf{Z}_i\}_{i=1}^m$ and train each epoch by iterating over the set of reference batches together with input batches of primary data points. This is similar in vein to training with stochastic gradient descent.

Each batch of reference points $\mathbf{Z}_i$ will make the neural network yield an output in space $\mathcal{Y}$. Evaluating all batches $\{\mathbf{Z}_i\}_{i=1}^m$ will in turn yield $m$ outputs, which can be used in different ways depending on the application, like the ensemble approach (Dietterich et al., 2002; Sagi & Rokach, 2018) in classification tasks.

In a nutshell, TAP integration without batch training will incur an additional memory cost of $\mathcal{O}(d_z n_z)$, and batch training reduces the memory cost to $\mathcal{O}(\frac{d_z n_z}{m})$. For reference, the additional memory required for a forward path in TAP integrated model (written in PyTorch) is around 1.3GB for one training data point and 100K reference data of dimension 98.

# 5 Numerical Experiments

## 5.1 Performance Evaluation

In this subsection, we evaluate TAP by plugging it into neural network classifiers. We show the effectiveness of TAP by comparing the performance of TAP-integrated networks with other variants. The simulations are implemented on three real-world datasets in different areas. Full implementation details for all experiments in this section can be found in the Appendix for reproducibility.

**Datasets:** To ensure a comprehensive evaluation of the performance of TAP integration, we select/create three real-world cross-modal datasets in three different areas. All datasets are open-access and can be found online. A detailed dataset and pre-processing description can be found in the Appendix.

- **Computer Vision:** We start with the MNIST dataset (MNIST) (Deng, 2012). We crop the upper half of all images as the primary modality $\mathbf{X}$ for digit prediction and use the lower half of all images as the reference modality $\mathbf{Z}$ (without labels and without pairing). This creates two data modes that have guaranteed complementary information while having different distributions. In model inference (testing), the test data points are upper-half images from the primary modality that are not present in the training set.

- **Healthcare:** We use the Activity dataset (Activity) (Mohino-Herranz et al., 2019), where the Electrodermal Activity (EDA) signals are the primary modality $\mathbf{X}$ for predicting the subject activity. Thoracic Electrica Bioimpedance (TEB) signals are used as the reference dataset $\mathbf{Z}$ (without labels and without pairing). In model inference (testing), the test data points are EDA signals from the primary modality that are not present in the training set.

- **Remote Sensing:** We also choose the Crop dataset (Crop) (Khosravi et al., 2018; Khosravi & Alavipanah, 2019), where the optical features $\mathbf{X}$ are used to predict the crop type, and the radar readings are used as reference dataset $\mathbf{Z}$ (without labels and without pairing). In model inference (testing), the test data points are optical features from the primary modality that are not present in the training set.

There is no overlapping instance among the training data (in space $\mathcal{X}$), reference data (in space $\mathcal{Z}$), and test data (in space $\mathcal{X}$).

**Models:** It is difficult to directly compare the effectiveness of TAP against existing state-of-the-art neural network architectures since this work is the first to propose cross-modal learning from a fully unlabeled data modality. However, we can show the effectiveness of TAP by carefully examining the performance difference across different variants of TAP.

Our **Baseline** competitor is a single-modal neural network without TAP integration at all. However, TAP integration will bring three changes to the baseline model, including depth increase, the addition of cross-attention structure, and the reference data $\mathbf{Z}$. To examine the contribution of these three aspects, we further propose three competitors as follows:

1. **FFN**: We replace TAP with a feedforward network to create the FFN variant. So, if FFN performs worse than TAP, we are able to factor out the impact of depth increase with TAP integration.

2. **Control Group**: We replace the real reference data $\mathbf{Z}$ in TAP with random noise of the same mean and variance. So, if Control Group performs worse than TAP, it suggests that the addition of a cross-attention structure is not the main reason for TAP to work, and using meaningful reference data (like $\mathbf{Z}$) is important.

3. **TAP w/o Batch**: We disregard batch training strategy by incorporating the whole reference dataset throughout the training process. So, if we observe TAP w/o Batch outperforms TAP, it would suggest the model is learning from the reference data, and more reference data will help.

We now highlight some important details here: First, the normalization parameter in the attention module of TAP is set to $\sqrt{d}(n_z/m)^{-1/d}$, such that it follows the bandwidth Assumption 1 while being close to the generally recommended normalization constant $\sqrt{d}$. Second, at each Monte-Carlo simulation, the set of training data, reference data, and evaluation data are shuffled while keeping the amount the same. The Control Group also generates a new set of random reference data at each Monte-Carlo simulation.

**Results:** The simulation results are shown in Figure 3. The error bars are calculated over 20 Monte-Carlo simulations to reflect the statistical significance of the results. As we can see, there is a consistent performance hierarchy among all the benchmark models throughout all datasets in different areas. First, we see that TAP integration always leads to a performance improvement compared to the baseline classifier. Second, we see that the FFN variant shows no performance advantage against the baseline model, which rules out the possibility that the depth increase in TAP is the major factor causing performance improvement. Third, we observe the worst generalization performance across all benchmark models for the Control Group. This

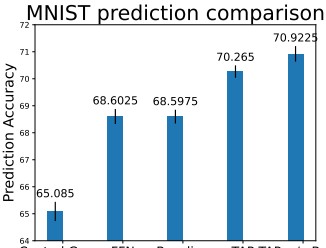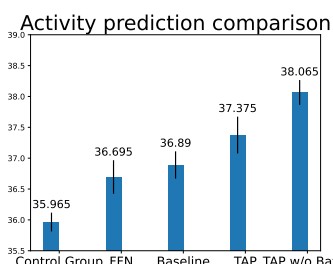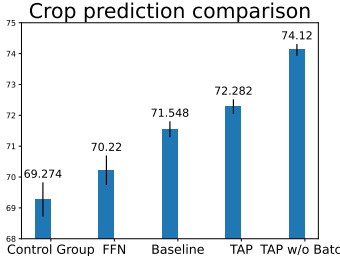

Figure 3: Simulation results on three real-world datasets. TAP integration shows a consistent performance advantage compared to other variants.

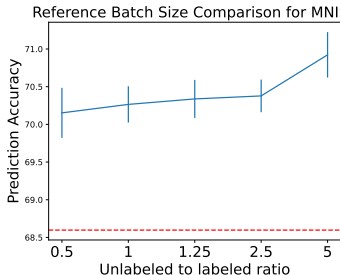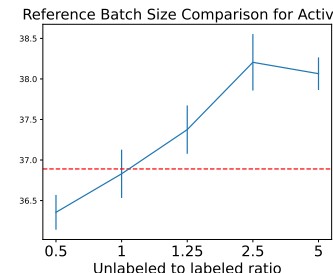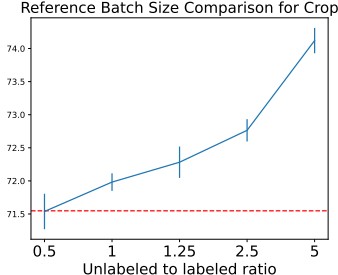

Figure 4: Reference batch size comparison on three real-world datasets. The generalization accuracy increases as the reference batch size becomes larger.

shows that feeding irrelevant information will exacerbate the generalization performance. Finally, we see that batch training and evaluation for TAP results in a slightly worse generalization performance compared to TAP w/o Batch, which trains directly with the whole reference dataset. Therefore, having more reference data indeed helps.

## 5.2 Ablation Study

In this subsection, we further conduct an ablation study for TAP to investigate its best configuration in practice. We discuss the batch size comparison and the choice of kernel function for TAP. Next, we look at the impact of nonlinear latent space transformation on TAP. Then, we examine the compatibility of TAP with large backbone feature extractors (for vision and language). Finally, we investigate shared space learning and dummy reference modality.

**Reference Batch Size Comparison:** To further evaluate the effect of batch training, we look at the performance of different reference batch sizes. The total number of reference data $n_z$ is five times of labeled data $n_x$. Figure 4 shows the prediction accuracy with respect to "unlabeled to labeled ratio", which is defined as the reference batch size divided by $n_x$.

The performance of the baseline model is shown as the horizontal dotted line. The standard errors are calculated over 10 Monte-Carlo simulations. We observe that the generalization performance of TAP improves as the reference batch size increases, even though the total number of reference data remains the same. This observation is consistent with the estimation error characterization in Theorem 1. In practice, this suggests that one can increase the reference batch size as much as possible until the memory limit is reached.

**Choice of Kernels:** Theorem 1 suggests that the estimation error diminishes as the number of reference data points goes to infinity. However, we observe that the error term is also related to $\mu_2(k)$, which is a constant determined by the choice of the kernel function. This connection is asymptotically negligible, but it might still be relevant in finite-sample cases. The choice of kernels in kernelized attention literature (Peng et al., 2020; Choromanski et al., 2020; Chen et al., 2021; Luo et al., 2021; Xiong et al., 2021) has been empirically studied, but these works have not suggested the theoretical intuition behind the kernel choices. Here, we

compare three kernels, namely the Gaussian kernel, Laplace kernel, and Inverse Multiquadric kernel. Notice that the $\mu_2(k)$ value is the smallest for the Gaussian kernel, larger for the Laplace kernel, and unbounded for the Inverse Multiquadric kernel.

Table 1: The prediction accuracy of TAP / TAP w/o Batch on three datasets. Gaussian kernel moderately outperforms the other two, and the Laplace kernel is slightly better than the Inverse Multiquadric kernel within the margin of error.

|  | Gaussian | Laplace | Inverse Multiquadric |
|---|---|---|---|
| MNIST | $70.3 \pm 0.25/\mathbf{70.9 \pm 0.33}$ | $69.9 \pm 0.21/70.1 \pm 0.23$ | $69.7 \pm 0.20/70.1 \pm 0.24$ |
| Activity | $37.4 \pm 0.25/38.1 \pm 0.19$ | $37.1 \pm 0.22/38.4 \pm 0.22$ | $37.3 \pm 0.28/38.5 \pm 0.16$ |
| Crop | $72.3 \pm 0.21/\mathbf{74.1 \pm 0.19}$ | $72.2 \pm 0.15/72.8 \pm 0.17$ | $71.9 \pm 0.14/72.5 \pm 0.25$ |

The results are shown in Table 1. The standard errors are calculated over 20 Monte-Carlo simulations. We observe that for TAP, the Gaussian kernel consistently outperforms the other two kernels. The general performance hierarchy is also consistent with the order of $\mu_2(k)$ value for three kernels. So, we recommend using the Gaussian kernel for the TAP integration.

**Nonlinear Latent Space Transformation:** We further investigate whether nonlinear latent space transformation will benefit TAP. We compare TAP with a variant where the latent space transformations $\phi_1, \phi_2, \phi_3$ are modeled with multi-layer perceptrons (MLP). The results are shown in Table 2.

Table 2: The prediction accuracy of TAP / TAP w/o Batch on three datasets. MLP variant shows little to no improvements compared to the linear version of TAP.

|  | Linear | MLP |
|---|---|---|
| MNIST | $70.3 \pm 0.25/70.9 \pm 0.22$ | $70.3 \pm 0.27/70.4 \pm 0.25$ |
| Activity | $37.4 \pm 0.25/38.1 \pm 0.19$ | $37.8 \pm 0.22/38.2 \pm 0.16$ |
| Crop | $72.3 \pm 0.21/74.1 \pm 0.19$ | $\mathbf{73.7 \pm 0.16}/74.4 \pm 0.26$ |

The standard errors are calculated over 10 Monte-Carlo simulations. We observe that MLP variants in general provide an accuracy that is no worse than the linear version of TAP. The only statistical advantage of MLP can be found in the Crop dataset, where the data are radar and optical images, which are information-rich and often require nonlinear mappings for feature extraction. We suggest using nonlinear latent space transformation, including backbone feature extractors for vision or language data, but adhering to linear transformation for tabular data for computational purposes.

**Backbone Compatibility:** We note that TAP considers cross-modal learning from a probability theory standpoint, which is domain agnostic. However, given the popularity of computer vision and large language models, it is appealing to see if TAP is compatible with pre-trained feature extractors, such as Convolutional Neural Networks (CNNs) for images and Transformers for language.

Table 3: The prediction accuracy (%) and F1 score comparison between baseline model and TAP integrated model. TAP integration shows a clear advantage in both accuracy and F1 score.

|  | Accuracy(%) | F1 Score |
|---|---|---|
| Baseline | $44.56 \pm 1.07$ | $0.453 \pm 0.005$ |
| TAP | $\mathbf{47.68 \pm 0.48}$ | $\mathbf{0.470 \pm 0.006}$ |

We carry out the test on a fourth dataset, Memotion 7K dataset (Sharma et al., 2020). Specifically, we focus on sentiment classification tasks with images as the primary modality and text as the secondary modality. We investigate a modified TAP integration as shown in Figure 5. We choose pre-trained EfficientNet-B0 (Tan & Le, 2019) as the image feature extractor and pre-trained distilled-RoBERTa (Sanh et al., 2019) as the text

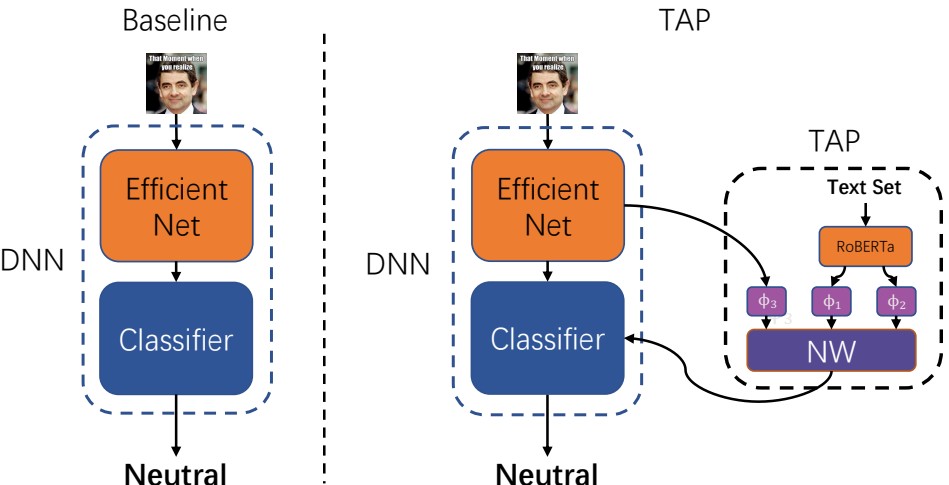

Figure 5: TAP integration with pre-trained feature extractors: The primary modality prediction model takes a meme image as input to predict the sentiment of the meme. A text set of batch size 100 is used as the reference secondary modality in TAP. The text data goes through pre-trained distilled-RoBERTa before being used in TAP.

feature extractor. The implementation details can be found in the Appendix. The test accuracy and F1 score are tabulated in Table 3. We observe that TAP integration improves both test accuracy and F1 score, which is the main objective for the imbalanced Memotion 7K dataset.

**Shared Space Learning:** The benefit of TAP relies on finding a shared information space $\mathcal{M}_1$, such that similar samples in the primary modality and reference modality are close to each other in the space $\mathcal{M}_1$ (i.e., after proper transformations). Although we have observed clear generalization advantage, it is still important to examine whether TAP is able to find such suitable space $\mathcal{M}_1$. To this end, we look at the kernel values in TAP after training for $\mathbf{x}$ and $\mathbf{z}$ with the same/different labels. The results are shown in Table 4.

Table 4: The kernel evaluations between data (one from the primary modality and one from the reference modality) that shares the same label against data with different labels.

| Dataset | Same Label | Different Label |
|---------|------------|-----------------|
| MNIST | $\mathbf{0.4008 \pm 0.0001}$ | $0.3945 \pm 0.0001$ |
| Activity | $\mathbf{0.7655 \pm 0.0004}$ | $0.7606 \pm 0.0003$ |
| Crop | $\mathbf{0.6424 \pm 0.0001}$ | $0.6398 \pm 0.0001$ |

The standard errors are calculated over all evaluation data. We observe that the kernel value $k(\mathbf{W}_q\mathbf{x}, \mathbf{W}_k\mathbf{z})$ is larger when $\mathbf{x}$ and $\mathbf{z}$ share the same label, which means TAP is able to find the appropriate shared space $\mathcal{M}_1$ without accessing label information for reference modality $\mathbf{Z}$. This observation helps explain why TAP can provide better generalization performance.

**Dummy Reference Modality:** To further verify the claim that the "extra information" in the reference modality improves learning, we carry out a simulation tricking TAP by creating dummy reference modalities for all previous three real-world datasets. To be specific, we create the reference modality in Section 5.1 by transforming the training data in the primary modality using randomly initialized feedforward neural networks (with the same output dimension as the reference data). For example, for Crop dataset, we randomly initialize a neural network with input dimension 76 (for optical features) and output dimension 98 (for radar features) to transform the primary modality training data into the reference dataset. Using this we have created a

"reference-like" dataset without any relevant cross-modal information. With 20 Monte-Carlo simulations, we get the performance results shown in Table 5.

Table 5: The performance comparison of the baseline model compared to TAP with dummy reference modality.

| Dataset | Baseline | TAP with dummy reference |
|---|---|---|
| MNIST | $69.91 \pm 0.37$ | $70.11 \pm 0.24$ |
| Activity | $36.99 \pm 0.26$ | $36.78 \pm 0.15$ |
| Crop | $70.52 \pm 0.42$ | $68.6 \pm 0.36$ |

We observe that TAP with "dummy" reference modality has no statistically significant performance gain over the baseline model if not being worse. This is intuitive as the embedded cross-modal information does not carry any extra information.

## 6 Conclusion and Discussion of Future Directions

This paper investigates a novel perspective on cross-modal learning, where the objective is to transfer knowledge from an unlabeled secondary modality to improve generalization in the primary modality. We showed that the missing information in the primary modality can be estimated using an NW kernel regression formulation. This perspective leads to the design of a cross-attention module in neural network integration, which we refer to as The Attention Patch (TAP). We demonstrated the effectiveness of TAP integration on four real-world cross-modal datasets in different domains, highlighting its performance advantage and domain robustness.

We hope that this work will inspire new perspectives on knowledge transfer from the data level for neural networks, showing how seemingly unusable data can be leveraged for improved generalization. However, our work can still be improved in the future in certain aspects. For example, having the secondary modality as keys and values requires large computation memory in the forward pass for large datasets, and there is also no guarantee that a shared space and an exclusive space exist between the two modalities. Therefore, there are several promising directions for future research. For example, developing approaches that can efficiently store and reference unlabeled modalities could significantly benefit the technique, given the wide availability of massive datasets. Additionally, a principled framework for selecting relevant reference datasets, as seen in cross-modal and semi-supervised learning, could help guide the data selection process.

## Acknowledgments

The authors gratefully acknowledge the support of NSF Award #2038625 as part of the NSF/DHS/DOT/NIH/USDA-NIFA Cyber-Physical Systems Program.

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

# A  Appendix

The organization of this Appendix is as follows:

- Subsection A.1 presents the experimental details, including dataset descriptions, preprocessing descriptions, and training details for reproducibility.

- In Subsection A.2, we provide the proof for all of the theoretical claims in the paper.

## A.1  Experimental Details

### A.1.1  Dataset

**Dataset Description:**

MNIST is a well-known dataset with 784 pixel value features. We crop the upper half of images as the primary modality, which corresponds to the first 392 features in the dataset, and the other 392 features form the lower half images. In this case, we know for sure that the two modalities are located in different spaces and carry complementary information.

Activity is an activity prediction dataset that predicts the status of a person through wearable physiological measurements. There are 151 features for Thoracic Electrical Bioimpedance (TEB) readings and 208 features for Electrodermal Activity (EDA) readings, including 104 for left arm readings and 104 for right arm readings. The objective is to predict the activity that the subject is going through, including neutral, mental, emotional, and physical activities.

Crop is a crop type prediction dataset that predicts the type of crop a land grows using satellite and radar readings. There are 76 optical features collected with RapidEye satellites and 98 radar features collected with the Unmanned Aerial Vehicle Synthetic Aperture Radar system. The labels contain seven crop types, including Corn, Peas, Canola, Soybeans, Oats, Wheat, and Broadleaf.

Memotion 7K is a meme image and text caption dataset. This dataset has several prediction objectives, and we implement sentiment prediction in our simulation. For each meme image and its text caption, the instance can be labeled as very negative, negative, neutral, positive, and very positive. We treat the image dataset as the primary modality and the unlabeled text caption dataset as the secondary modality.

**Data Pre-processing:**

Similar to semi-supervised learning, the motivation behind utilizing unlabeled data points is the limited availability of labeled data. So, we randomly sample 200 data points in the primary modality to serve as the training data for each dataset. We further randomly sample 1000 data points in the secondary modality to serve as the cross-modal reference data. For MNIST, it means 200 upper half images as primary modality training data and 1000 lower half images as reference data. All the remaining data points will be the evaluation data.

The memotion 7K dataset requires more training data to learn a model. So, we randomly sample 5000 data as the training set, 1000 data as the reference data, and the rest of data points are the evaluation data.

### A.1.2  Performance Evaluation

**Parameter Setting:**

In the performance evaluation, the reference batch size for TAP is chosen as 250, which is 1.25 times the training data. The training data batch size is set to 100. As shown in the reference batch size comparison, 250 is not the best-performing choice, but it always yields better generalization accuracy on first three datasets.

The backbone neural network structure for all three datasets is a two-hidden-layer neural network with 64 hidden neurons at each layer. The activation function is ReLU with a dropout rate of 0.5. Layer normalization is implemented after each hidden layer. TAP is integrated between the two hidden layers. For TAP, TAP

w/o Batch, FFN, and Control Group, the second hidden layer still has 64 neurons, except the linear layer that takes input from $\mathbb{R}^{128}$ and outputs $\mathbb{R}^{64}$.

The hidden dimension in TAP is set to 64, and the output dimension is also 64. FNN replaces the attention module in TAP with a linear layer from $\mathbb{R}^{64}$ to $\mathbb{R}^{64}$. Layer normalization is implemented in TAP (including TAP w/o Batch and Control Group) and FNN as well. The normalization parameter in the attention module of TAP is set to $\sqrt{d}(n_z/m)^{-1/d}$, such that it follows the bandwidth Assumption 1 while being close to the generally recommended normalization constant $\sqrt{d}$.

At each Monte-Carlo Simulation, the set of training data, reference data, and evaluation data are shuffled while keeping the amount the same. The Control Group also generates a new set of random reference data at each Monte-Carlo simulation.

**Network Training:**

All benchmark variant models share a similar backbone structure; therefore, the training process is also similar. All models are trained with the cross-entropy loss using the Adam optimizer with a fixed learning rate of 0.0001. All models are trained for 1000 epochs (8000 in the Crop dataset) except for TAP. Since TAP updates the model several times within one epoch, the total training epochs for TAP is reduced to $1000/m$, where $m$ is the number of reference batches. For example, in the case of performance evaluation where the reference batch size is 250, there are four batches of reference data, and the total training epochs is 250 for TAP. The prediction accuracy is calculated as the lowest five-moving-average of validation accuracy throughout the training process, which is a common practice in neural network performance evaluation.

The batch evaluation of TAP is done by having each batch of reference data generate a prediction class and finding the majority vote among all batches just like ensemble methods.

### A.1.3 Ablation Study

**Reference Batch Size Comparison:** The experimental settings in this section of the simulation are very much identical to the previous section. The reference batch sizes are chosen as $100, 200, 250, 500, 1000$, which corresponds to the unlabeled to labeled ratios of $1/2, 1, 5/4, 5/2, 5$, respectively. The corresponding training epochs are $100, 200, 250, 500, 1000$, respectively. For the Crop dataset, all training epochs are multiplied by 8 since it takes longer to train.

**Choice of Kernels:** This section adds two kernel variants of TAP integration. The only difference between standard TAP and these two variants is the kernel function in Equation (9). The Laplace kernel function is set to

$$k(\mathbf{x}, \mathbf{y}) = e^{-\frac{\|\mathbf{x}-\mathbf{y}\|_1}{h^2}}, \tag{10}$$

where $\|\cdot\|_1$ denotes the $\ell_1$-norm, and the normalizing factor $h$ is chosen the same as the Gaussian kernel bandwidth in TAP. The Inverse Multiquadric kernel is set to

$$k(\mathbf{x}, \mathbf{y}) = \frac{1}{\sqrt{\|\mathbf{x} - \mathbf{y}\|_2 + 1}}. \tag{11}$$

All the other training settings are kept the same as the performance evaluation section for first three datasets.

**Nonlinear Latent Space Transformations:** This section adds the MLP variant of TAP integration. In this variant, linear transformations are replaced with a two-hidden-layer neural network with a ReLU activation function. All the other training settings are kept the same as the performance evaluation section for first three datasets.

**Backbone Compatibility:** In this simulation, the classifier appended to feature extractors is a two-hidden-layer neural network with 16 neurons at each layer. The learning rate is set to 0.00002 for both baseline and TAP integration. Each model is trained for 20 epochs, where we observe the validation accuracy stabilizes. The reported accuracy is the average of the last five validation accuracy values.

**Shared Space Learning:** The model and learning process are the same as experiments in performance evaluation. We use the trained-model with the highest accuracy in the last Monte-Carlo simulation to evaluate the kernel values.

### A.2 Theory of TAP

### A.2.1 Proof of Proposition 1

We rewrite the conditional expectation in Equation (1) as follows

$$
\begin{aligned}
\mathbb{E}(\mathbf{m}_2|\mathbf{m}_1) &= \int \mathbf{m}_2 \frac{p_1(\mathbf{m}_2, \mathbf{m}_1)}{p_2(\mathbf{m}_1)} d\mathbf{m}_2 \\
&= \int \phi_2(\mathbf{z}) \frac{p_1(\mathbf{m}_2, \mathbf{m}_1)}{p_2(\mathbf{m}_1)} d\phi_2(\mathbf{z}).
\end{aligned}
\tag{12}
$$

Next, plugging the density estimations in Equation (2) back into the above equation, we have the following

$$
\begin{aligned}
\mathbb{E}(\mathbf{m}_2|\mathbf{m}_1) &\approx \int \phi_2(\mathbf{z}) \frac{\hat{p}_1(\mathbf{m}_2, \mathbf{m}_1)}{\hat{p}_2(\mathbf{m}_1)} d\phi_2(\mathbf{z}) \\
&= \sum_{i=1}^{n_z} \frac{\int \phi_2(\mathbf{z}) k(\phi_3(\mathbf{x}), \phi_1(\mathbf{z}_i)) k_1(\phi_2(\mathbf{z}), \phi_2(\mathbf{z}_i)) d\phi_2(\mathbf{z})}{\sum_{j=1}^{n_z} k(\phi_3(\mathbf{x}), \phi_1(\mathbf{z}_j))} \\
&= \sum_{i=1}^{n_z} \frac{k(\phi_3(\mathbf{x}), \phi_1(\mathbf{z}_i)) \int \phi_2(\mathbf{z}) k_1(\phi_2(\mathbf{z}), \phi_2(\mathbf{z}_i)) d\phi_2(\mathbf{z})}{\sum_{j=1}^{n_z} k(\phi_3(\mathbf{x}), \phi_1(\mathbf{z}_j))}, \\
&= \sum_{i=1}^{n_z} \frac{k(\phi_3(\mathbf{x}), \phi_1(\mathbf{z}_i)) \phi_2(\mathbf{z}_i)}{\sum_{j=1}^{n_z} k(\phi_3(\mathbf{x}), \phi_1(\mathbf{z}_j))},
\end{aligned}
\tag{13}
$$

where the last line simply follows from the fact that kernel function $k_1(\cdot, \boldsymbol{\mu})$ has a mean $\boldsymbol{\mu}$. The proof is complete.

### A.2.2 Lemma

**Lemma 1.** *Under Assumptions 1 and 2, a kernel density estimator*

$$
\hat{p}(\mathbf{x}) = \frac{1}{n} \sum_{i=1}^{n} k_{\mathbf{H}}(\mathbf{x}, \mathbf{x}_i),
\tag{14}
$$

*is consistent, i.e., it converges in probability to the true density function $p(\mathbf{x})$,*

$$
\hat{p}(\mathbf{x}) \xrightarrow{p} p(\mathbf{x}).
\tag{15}
$$

This is a well-known result that can be found in the existing literature (Devroye & Wagner, 1979).

### A.2.3 Proof of Theorem 1

Throughout the proof, all integrals are Riemann integral $\int_{\mathbb{R}^d}$, and we omit the subscript for simplicity. Since we only consider shift-invariant kernels, we will use $k_{\mathbf{H}}(\mathbf{x} - \mathbf{x}_i)$ to represent an isotropic kernel function with bandwidth $\mathbf{H} = h_n \mathbf{I}$, where $k_{\mathbf{H}}(\mathbf{x} - \mathbf{x}_i) = \frac{1}{|\mathbf{H}|} k(\frac{\mathbf{x} - \mathbf{x}_i}{|\mathbf{H}|})$ for a standard shift-invariant kernel function $k(\cdot)$. The determinant of the bandwidth matrix by definition is $|\mathbf{H}| = h_n^d$. We will further omit the subscripts of $h$ for simplicity.

Consider the $j$-th element in $\mathbf{z}_i$, which we simply write as $z^{(j)}$. Then, we have the following

$$
\begin{aligned}
\hat{f}^{(j)}(\mathbf{x}) &= \frac{\frac{1}{n}\sum_{i=1}^n k_{\mathbf{H}}(\mathbf{x}-\mathbf{x}_i)z^{(j)}}{\hat{p}(\mathbf{x})} \\
&= \frac{\frac{1}{n}\sum_{i=1}^n k_{\mathbf{H}}(\mathbf{x}-\mathbf{x}_i)(f^{(j)}(\mathbf{x})+f^{(j)}(\mathbf{x}_i)-f^{(j)}(\mathbf{x})+\epsilon_i^{(j)})}{\hat{p}(\mathbf{x})} \\
&= f^{(j)}(\mathbf{x}) + \frac{\frac{1}{n}\sum_{i=1}^n k_{\mathbf{H}}(\mathbf{x}-\mathbf{x}_i)(f^{(j)}(\mathbf{x}_i)-f^{(j)}(\mathbf{x}))}{\hat{p}(\mathbf{x})} + \frac{\frac{1}{n}\sum_{i=1}^n k_{\mathbf{H}}(\mathbf{x}-\mathbf{x}_i)\epsilon_i^{(j)}}{\hat{p}(\mathbf{x})} \\
&\triangleq f^{(j)}(\mathbf{x}) + \frac{B(\mathbf{x})}{\hat{p}(\mathbf{x})} + \frac{V(\mathbf{x})}{\hat{p}(\mathbf{x})},
\end{aligned}
\tag{16}
$$

where we theoretically characterize the properties of $B(\mathbf{x})$ and $V(\mathbf{x})$. We refer to the $i$-th term in the summation $B(\mathbf{x})$ and $V(\mathbf{x})$ as $b_i(\mathbf{x})$ and $v_i(\mathbf{x})$, and the subscripts will be omitted from now on for simplicity.

We first look at the term $v(\mathbf{x})$, and it is easy to see that $\mathbb{E}[v(\mathbf{x})] = 0$ due to our assumption on the noise term $\boldsymbol{\epsilon}$. For the variance, we have the following asymptotic equality

$$
\begin{aligned}
\mathbb{E}[|\mathbf{H}|v^2(\mathbf{x})] &= \mathbb{E}[\epsilon^2]|\mathbf{H}|\int k_{\mathbf{H}}^2(\mathbf{x}-\mathbf{y})p(\mathbf{y})d\mathbf{y} \\
&= \frac{\sigma^2}{|\mathbf{H}|}\int k^2(\frac{\mathbf{x}-\mathbf{y}}{|\mathbf{H}|})p(\mathbf{y})d\mathbf{y} \\
&= \sigma^2\int k^2(\mathbf{z})p(\mathbf{x}-|\mathbf{H}|\mathbf{z})d\mathbf{z} \\
&= \sigma^2\int k^2(\mathbf{z})(p(\mathbf{x})+o(1))d\mathbf{z} \\
&\asymp R(k)p(\mathbf{x})\sigma^2,
\end{aligned}
\tag{17}
$$

where the last line follows from omitting the higher order term. Since $V(\mathbf{x})$ is the sample mean of random variables $v(\mathbf{x})$, we can apply the central limit theorem to show that

$$
\sqrt{nh^d}V(\mathbf{x}) \xrightarrow{d} \mathcal{N}(0, R(k)p(\mathbf{x})\sigma^2).
\tag{18}
$$

Since $\hat{p}(\mathbf{x}) \xrightarrow{p} p(\mathbf{x})$ following Lemma 1, we can make the claim following Slutsky's theorem

$$
\frac{\sqrt{nh^d}V(\mathbf{x})}{\hat{p}(\mathbf{x})} \xrightarrow{d} \mathcal{N}(0, \frac{R(k)\sigma^2}{p(\mathbf{x})}).
\tag{19}
$$

Then, let us look at the term $b(\mathbf{x})$. We have the following

$$
\begin{aligned}
\mathbb{E}[b(\mathbf{x})] =& \frac{1}{|\mathbf{H}|}\mathbb{E}\Big[k(\frac{\mathbf{x}_i - \mathbf{x}}{|\mathbf{H}|})(f^{(j)}(\mathbf{x}_i) - f^{(j)}(\mathbf{x}))\Big] \\
=& \frac{1}{|\mathbf{H}|}\int k(\frac{\mathbf{y} - \mathbf{x}}{|\mathbf{H}|})(f^{(j)}(\mathbf{y}) - f^{(j)}(\mathbf{x}))p(\mathbf{y})d\mathbf{y} \\
=& \int k(\mathbf{z})(f^{(j)}(\mathbf{x} + |\mathbf{H}|\mathbf{z}) - f^{(j)}(\mathbf{x}))p(\mathbf{x} + |\mathbf{H}|\mathbf{z})d\mathbf{z} \\
\asymp& \int k(\mathbf{z})\Big((|\mathbf{H}|\mathbf{z})^\top\nabla f^{(j)}(\mathbf{x}) + \frac{1}{2}(|\mathbf{H}|\mathbf{z})^\top\nabla^2 f^{(j)}(\mathbf{x})(|\mathbf{H}|\mathbf{z})\Big)\Big(p(\mathbf{x}) + (|\mathbf{H}|\mathbf{z})^\top\nabla p(\mathbf{x})\Big)d\mathbf{z} \\
\asymp& |\mathbf{H}|\int k(\mathbf{z})\mathbf{z}^\top\nabla f^{(j)}(\mathbf{x})p(\mathbf{x})d\mathbf{z} + |\mathbf{H}|^2\mathrm{Tr}\left[\int k(\mathbf{z})p(\mathbf{x})\frac{1}{2}\nabla^2 f^{(j)}(\mathbf{x})\mathbf{z}\mathbf{z}^\top d\mathbf{z}\right] \\
& + |\mathbf{H}|^2\mathrm{Tr}\left[\int k(\mathbf{z})\nabla f^{(j)}(\mathbf{x})\nabla p(\mathbf{x})^\top\mathbf{z}\mathbf{z}^\top d\mathbf{z}\right] \\
=& |\mathbf{H}|^2\mu_2(k)\Big(\frac{1}{2}p(\mathbf{x})\mathrm{Tr}\left[\nabla^2 f^{(j)}(\mathbf{x})\right] + \mathrm{Tr}\left[\nabla f^{(j)}(\mathbf{x})\nabla p(\mathbf{x})^\top\right]\Big) \\
\triangleq& |\mathbf{H}|^2\mu_2(k)\mathbf{\Psi}^{(j)}(\mathbf{x}) = h^{2d}\mu_2(k)\mathbf{\Psi}^{(j)}(\mathbf{x}).
\end{aligned}
\tag{20}
$$

Any term that is $o(|\mathbf{H}|^2)$ is omitted as it asymptotically vanishes.

For the term $\mathbb{E}[b^2(\mathbf{x})]$, we use a similar expansion by omitting terms of $o(|\mathbf{H}|^2)$ as follows

$$
\begin{aligned}
\mathbb{E}[b^2(\mathbf{x})] =& \frac{1}{|\mathbf{H}|^2}\mathbb{E}\Big[k^2(\frac{\mathbf{x}_i - \mathbf{x}}{|\mathbf{H}|})(f^{(j)}(\mathbf{x}_i) - f^{(j)}(\mathbf{x}))^2\Big] \\
=& \frac{1}{|\mathbf{H}|^2}\int k^2(\frac{\mathbf{y} - \mathbf{x}}{|\mathbf{H}|})(f^{(j)}(\mathbf{y}) - f^{(j)}(\mathbf{x}))^2 p(\mathbf{y})d\mathbf{y} \\
=& \frac{1}{|\mathbf{H}|}\int k^2(\mathbf{z})(f^{(j)}(\mathbf{x} + |\mathbf{H}|\mathbf{z}) - f^{(j)}(\mathbf{x}))^2 p(\mathbf{x} + |\mathbf{H}|\mathbf{z})d\mathbf{z} \\
\asymp& \frac{1}{|\mathbf{H}|}\int k^2(\mathbf{z})((|\mathbf{H}|\mathbf{z})^\top\nabla f^{(j)}(\mathbf{x}))^2 p(\mathbf{x})d\mathbf{z} \\
=& O\Big(|\mathbf{H}|\nabla f^{(j)}(\mathbf{x})^\top\nabla f^{(j)}(\mathbf{x})p(\mathbf{x})\mu_2(k)\Big).
\end{aligned}
\tag{21}
$$

This suggests that the variance of $b(\mathbf{x})$ is of a higher order than the variance of $\sqrt{|\mathbf{H}|}v(\mathbf{x})$. Thus, we know

$$
\sqrt{nh^d}(B(\mathbf{x}) - h^{2d}\mu_2(k)\mathbf{\Psi}^{(j)}(\mathbf{x})) \xrightarrow{p} 0.
\tag{22}
$$

Combining with the convergence in probability for density estimator $\hat{p}(\mathbf{x})$, we have

$$
\sqrt{nh^d}(\frac{B(\mathbf{x})}{\hat{p}(\mathbf{x})} - \frac{h^{2d}\mu_2(k)\mathbf{\Psi}^{(j)}(\mathbf{x})}{p(\mathbf{x})}) \xrightarrow{p} 0.
\tag{23}
$$

Finally, we conclude that

$$
\begin{aligned}
& \sqrt{nh^d}(\mathbf{e}(\mathbf{x}) - \frac{h^{2d}\mu_2(k)\mathbf{\Psi}(\mathbf{x})}{p(\mathbf{x})}) \\
=& \sqrt{nh^d}(\frac{\mathbf{B}(\mathbf{x})}{\hat{p}(\mathbf{x})} - \frac{h^{2d}\mu_2(k)\mathbf{\Psi}(\mathbf{x})}{p(\mathbf{x})} + \frac{\mathbf{V}(\mathbf{x})}{\hat{p}(\mathbf{x})}) \xrightarrow{d} \mathcal{N}(\mathbf{0}, \frac{R(k)\sigma^2}{p(\mathbf{x})}\mathbf{I}),
\end{aligned}
\tag{24}
$$

where $\mathbf{B}(\mathbf{x})$ and $\mathbf{V}(\mathbf{x})$ are the vectorization of $B(\mathbf{x})$ and $V(\mathbf{x})$, respectively. The vectorization follows the isotropic assumption on noise $\boldsymbol{\epsilon}$.

The proof is complete.

