# OpenReview forum: "TAP: The Attention Patch for Cross-Modal Knowledge Transfer from Unlabeled Modality"
_TMLR — Accepted by TMLR_

### Review · Reviewer_Yp98 · 2024-02-03

**Summary Of Contributions:**

This paper studies semi-supervised learning when dealing with labeled primary data and unlabeled secondary data of different modalities. To enhance supervised learning on the labeled data, the paper introduces a cross-modal learning framework. The proposed approach involves estimating missing information through kernel density estimation, and further developing a kernelized cross-attention module (TAP) that could be inserted into the existing models. The study includes experiments conducted on several real-world datasets.

**Audience:**

Yes

**Broader Impact Concerns:**

/

**Claims And Evidence:**

Yes

**Requested Changes:**

see weakness

**Strengths And Weaknesses:**

Strengths:
1. The writing is clear and easy to follow. Fig. 1 illustrates the studied task well.
2. The idea of estimating the missing information using kernel density estimators is new to me and building cross-attention for better integrating the two different modalities sounds reasonable.
3. Experiments demonstrate the effectiveness of the proposed method on its formed datasets.

Weakness:
1. Important related works are missing. The claim that the problem is the first exploration of combining cross-modal learning and semi-supervised learning is overstated. Cross-domain few-shot learning (CD-FSL) [1,2] explores a similar setting, learning from labeled source data (primary) and unlabeled, unpaired target data (secondary) of different domains. While specific settings may differ, the assertion of being the first is challenged.
2. The testing data (T) should be defined and formulated, considering the supposed importance of domain distances between primary training data (X), secondary training data (Z), and testing data (T). The introduction of T in Sec. 5.1 might not be the most effective choice.
3. The constructed datasets for experiments lack convincing modal differences. The paper emphasizes tackling datasets of different modalities, but the constructed datasets do not seem to meet this requirement. For example, the use of the upper half and lower half of MNIST images as X and Z, respectively, raises questions about modality differences.
4. Discrepancies exist in the mention of datasets. The abstract and introduction state experiments on four datasets, while the experiments section includes only three.

[1] Self-training for few-shot transfer across extreme task differences. ICLR. 2021.
[2] Dynamic distillation network for cross-domain few-shot recognition with unlabeled data. NeurIPS. 2021.

---

> ### Author Response · Authors · 2024-04-05
> **Answers to the weaknesses**
>
> **Q1**:
> We would like to thank the reviewer for providing the reference papers. We did not intend to claim that we are the first work that considers intersection of semi-supervised and cross-modal learning. Instead, we made it specific that our work considers such intersection when the secondary modality is **unlabeled** and **unpaired with the primary modality**.
>
> We agree that the two provided references could also be considered at the intersection of cross-modal learning and semi-supervised learning. Upon careful reading of these papers, both references propose a two-step meta-learning approach with a focus on improving the few-shot learning performance for data in the same domain/modality as the unlabeled portion of the meta-dataset. We note that this approach can be regarded as an extension of co-learning into semi-supervised learning, where one modality is fully labeled, and another modality is partially labeled (unlabeled in meta dataset, few-shot labeled in actual learning). Our target, on the other hand, is a one-step learning approach that allows one modality to be **fully unlabeled** throughout the learning process.
>
> Please see our first contribution bullet. To address the contribution statement issue, we have rephrased the contribution by narrowing down the scope.
>
> We have also added the references mentioned by the reviewer to the literature review on non-parallel co-learning paragraph in Section 2.1.
>
> **Q2**:
> We agree with the reviewer and added the test data description in Section 5.1. In a nutshell, the test data is a single modality data that comes from the same space as the primary modality X, and our goal is to predict the label for that data. We hope that this modification will help clarify the test data setting.
>
> **Q3**:
> We agree that the half-MNIST dataset is not generally regarded as a cross-modal dataset. However, in the created dataset (upper and lower half of MNIST images), the two “modalities” have different probability distributions (difference in the sense of either measure or support). The purpose of creating this dataset is to study a case that exactly follows Figure 1(a), where there exists guaranteed mutual information, and there is also exclusive information contained in each part of data. Therefore, the MNIST data serves as a proof of concept. On the other hand, we believe that Activity, Crop, and Memotion 7K datasets  are considered standard cross-modal datasets. For these datasets, we have clearly mentioned the data modes. For example, in Activity data we have Electrodermal Activity (EDA) signals (primary) vs. Thoracic Electrica Bioimpedance (TEB) signals (secondary).
>
> **Q4**:
> We use three datasets (MNIST, Activity, Crop) for the first part of experiments. Then, the fourth dataset (Memotion 7K dataset) appears in the backbone compatibility test. To avoid any potential confusions, we double checked the manuscript and rephrased the third contribution bullet and abstract accordingly.

---

### Review · Reviewer_6oEH · 2024-02-06

**Summary Of Contributions:**

The paper presents a plug-in module called The Attention Patch (TAP) to leverage complementary information from an unlabeled modality for semi-supervised cross-modal learning. Specifically, the authors propose TAP as a kernelized cross-attention module between the labeled and unlabeled modalities, and formulate the task as a NW Kernel Regression. The key insight is that TAP can provide meaningful signals as the data scales up. Empirical results on 4 datasets shows that TAP consistently improves the performance on classification tasks.

**Audience:**

Yes

**Broader Impact Concerns:**

No significant broader impact concerns.

**Claims And Evidence:**

No

**Requested Changes:**

* Please refer to the weaknesses above (for convincing results, use a more suited architecture for each task, evaluate using transformer backbones, and verify if TAP functions as expected using qualitative visualizations).

* According to Assumption 2, "the true density function p(x) is differentiable". Could the authors clarify if this assumption imposes limitations on the kinds of datasets / tasks / input modalities?

**Strengths And Weaknesses:**

Strengths:
* The manuscript is easy to read. Fig. 1 is clear and succinctly describes the problem setting.
* The authors propose an important and practical setting where the reference dataset is unlabeled, is unpaired, and, can come from a different modality.
* The paper introduces a theoretical argument to the success of the attention patch which helps the reader develop an intuition in this area.
* The comparison of TAP with batch and without batch is quite interesting and shows promising generalization given unlabeled data at scale.

Weaknesses:
* The evaluation seems unconvincing to support the claims:
1. Suppl. Sec. A.1.2 describes "All benchmark variant models share a similar backbone structure", whereas the tasks are quite different - "MNIST" classifies images whereas "Activity" classifies signals. The authors use a trivial architecture (MLP) and it is unclear if TAP would show performance gains when using a more suitable model architecture for each task.
2. TAP is similar in nature to well-studied multi-head attention and therefore it is imperative that the authors evaluate a transformer backbone to demonstrate the gains of TAP over vanilla transformers, however such a comparison is missing.
3. The kernel evaluations in Table 4 only show a weak difference (about 0.005) between the two cases which raises questions regarding the effectiveness of TAP (could this difference be negligible when using a more suitable architecture for the tasks?). Moreover, the ablations in Table 1 for TAP do not seem to be statistically significant (accuracy scores are within standard deviation).
4. The most practical use-case (integrating TAP in pretrained feature extractors; Fig. 5) is under-studied -- Table 3 provides no insight into how TAP functions when using cross-modal data such as image and text. For instance, it is important to visualize what text caption is retrieved for a given image, to verify TAP indeed works in retrieving complementary information from the reference modality (similarly, for MNIST, given the upper half of the image, verify whether the lower half is retrieved from TAP).
* Key details missing: Fig. 2 caption states "the output of TAP will be fed to the next layer together with TAP input" - is this a concatenation operation, or are the two inputs added? Further, could the authors clarify if backpropagation happens through the TAP module?

---

> ### Author Response · Authors · 2024-04-05
> **Answers to weaknesses 1 and 2**
>
> **Q1**:
> We thank the reviewer for raising this point. Activity and Crop datasets are commonly modeled with MLPs:
>
> [1] Silva, A. A., Xavier, A. S., Macêdo, D., Zanchettin, C., & Oliveira, A. L. (2022, July). An Adapted GRASP Approach for Hyperparameter Search on Deep Networks Applied to Tabular Data. In 2022 International Joint Conference on Neural Networks (IJCNN) (pp. 1-8). IEEE.
>
> [2] Orynbaikyzy, A., Gessner, U., & Conrad, C. (2019). Crop type classification using a combination of optical and radar remote sensing data: A review. international journal of remote sensing, 40(17), 6553-6595.
>
> Moreover, MLP is also one of the top benchmark architectures for MNIST dataset:
>
> [3] Wan, L., Zeiler, M., Zhang, S., Le Cun, Y., & Fergus, R. (2013, May). Regularization of neural networks using dropconnect. In International conference on machine learning (pp. 1058-1066). PMLR.
>
> We agree that certain high-dimensional data (like image and text data) need proper backbone for meaningful examination, and this is exactly why we have provided the backbone compatibility test in Section 5.2, where the image modality is paired with CNN-based efficient net, and test data is paired with Transformer-based distilled RoBERTa. We observed similar performance advantage for TAP integration in this scenario as well.
>
> **Q2**:
> We would appreciate a clarification for this question. If the reviewer refers to comparison against the standard Transformer architecture as in
>
> “Vaswani, A., Shazeer, N., Parmar, N., Uszkoreit, J., Jones, L., Gomez, A. N., ... & Polosukhin, I. (2017). Attention is all you need. Advances in neural information processing systems, 30.”
>
> we want to highlight that TAP focuses on cross-modal learning with unlabeled reference modality, which is different from the target of vanilla Transformer.
>
> If the reviewer refers to the compatibility of TAP with transformer backbone (or attention-based feature extractor in particular) for supervised learning tasks, then we believe the simulation with distilled-RoBERTa gives answer to the question.

---

> ### Author Response · Authors · 2024-04-05
> **Answers to weaknesses 3 and 4**
>
> **Q3**:
> In Table 4, the kernel evaluation difference is small in the sense of absolute value; However, it is still statistically much larger than the standard error (50 times). Therefore, the difference across kernel evaluations is statistically significant. We believe this is more of an issue of normalization where embedded vectors are normalized to the same scale.
>
> We agree that in Table 1 the differences are not statistically significant. However, we are not comparing TAP with other methods in that experiment. Instead, we are just highlighting the role of $\mu_2(k)$ for various choices of kernels.
>
> **Q4**:
> Thank you for this comment. We would like to highlight the difference between **cross-modal retrieval** and the problem we consider in this paper. In cross-modal retrieval, the main objective is to retrieve or reconstruct X from Z and/or Z from X, so there are only two spaces involved, while TAP focuses on a supervised learning objective where we use data in X to predict labels Y with the help of unlabeled, unpaired data from Z. For example, given that data modes are unpaired, we cannot even observe what “lower half” digit is recovered for a particular “upper half” in MNIST.
>
> The reviewer certainly mentions a relevant literature and we acknowledged the key differences between the target task of TAP and cross-modal retrieval in alignment requirement in Section 2.1 (first paragraph).
>
> We think the concept of TAP (based on density estimation) could potentially contribute to instance-wise non-aligned cross-modal retrieval, but there is still much more work to be done for TAP to be integrated to cross-modal retrieval tasks.

---

> ### Author Response · Authors · 2024-04-05
> **Answers to other comments**
>
> >Key details missing: Fig. 2 caption states "the output of TAP will be fed to the next layer together with TAP input" - is this a concatenation operation, or are the two inputs added? Further, could the authors clarify if backpropagation happens through the TAP module?
>
> The output of TAP is concatenated with the last layer’s output rather than a residual-like connection, and back-propagation indeed happens to the weights of TAP as stated in Corollary 1 “The parameters $W_q, W_k, W_v$ are learned in parallel with the original neural network”. To elaborate on the choice of concatenation, we have tested both while observing that residual-like connections do not bring statistically significant improvements to the generalization. Following reviewer’s suggestion, we changed the caption to “the output of TAP will be concatenated with TAP input and fed to the next layer” in the revised manuscript.
>
> >According to Assumption 2, "the true density function p(x) is differentiable". Could the authors clarify if this assumption imposes limitations on the kinds of datasets / tasks / input modalities?
>
> The assumption of $p(x)$ being differentiable is a standard assumption in the statistics literature (Wand, M. P., & Jones, M. C. (1994). Kernel smoothing. CRC press.), imposed to make sure our density function can be approximated with Taylor series; this assumption is essential for the convergence of our theorem. However, we did not observe evidence of any limitations imposed by this theorem in practice, as simulation has demonstrated good performance on discrete density functions (image data with countable RGB or gray scale spectrum) as well.

---

### Review · Reviewer_XGTR · 2024-03-22

**Summary Of Contributions:**

- The paper attacks the problem of cross-modal learning.
- The authors propose a cross-modal mechanism based on the Nadaraya-Watson kernel regression.
- In the experiments, they provide evidence in favor of their approach.

**Audience:**

Yes

**Broader Impact Concerns:**

No.

**Claims And Evidence:**

Yes

**Requested Changes:**

- Please provide some complexity analysis. It is very important for broader applicability to indicate the additional complexity of a DNN w/ TAP compared to a DNN w/o TAP, e.g., in terms of the big-O notation of extra calculations. Moreover, what is the number of parameters of the DNN w/ TAP compared to its version w/o TAP.
- Figure 3: Please enlarge the font size.
- Figure 4: Please remove the yellow background and enlarge the font.
- Please consider adding a new experiment as described in the weaknesses.

**Strengths And Weaknesses:**

Strengths:
- The paper tackles an interesting problem.
- The paper is rather well-written.
- The proposed methodology seems reasonable.

Weaknesses:
- I miss a complexity analysis. At the moment, it is hard for me to assess the difference between a DNN w/ TAP and a DNN w/o TAP in terms of, e.g., the number of weights.
- I presume the authors will hate me for this comment but it would be interesting to consider a case when the "extra" information z comes from the prime modality and is paired with an object x from the primary modality. Then it would be interesting to see how TAP deals in such a scenario. In other words, this experiment would weaken or strengthen the paper's first contribution.

---

> ### Author Response · Authors · 2024-04-05
> **Answers to the requested changes**
>
> **Q1**: We thank the reviewer for the suggestion, and we expanded upon the brief discussion of the memory complexity in the reference batch size comparison section (just before Section 5). To summarize, the total number of weights introduced by TAP integration scales linearly with respect to the reference dimension $d_z$ and the reference data batch size $n_z/m $; it is thus $O(d_z n_z/m)$. For example, in practice, the forward path (written in PyTorch) incurs an additional 1.3Gb memory size for a reference data dimension of $98$ and a reference batch size of $100K$.
>
> **Q2 and Q3**: We have made changes according to reviewer’s suggestions.
>
> **Q4**: We thank the reviewer for suggesting this interesting experiment. We added the simulation suggested by the reviewer as **dummy reference modality** in the ablation study section (Sec 5.2). To summarize, we observed no statistical advantages in TAP integration with a reference modality that is a transformation of the primary modality. Intuitively, this makes sense as when there is no relevant extra information in the reference modality, there is no incentive in using TAP, as all the required information is encapsulated in the primary modality. Please see Table 5 in the revised manuscript for the results.

---

### Decision · Action_Editor_jSjX · 2024-05-25

**Recommendation:** Accept as is

**Comment:**

All major concerns of the reviewers have been addressed. Two of the reviewers recommend leaning to accept, but neither is very excited with the overall work.

**Audience:**

Yes, the proposed solution and theoretical analysis is of interest to the general TMLR community.

**Claims And Evidence:**

The paper examines the task of semi-supervised cross--modal learning problem and presents Attention Patch (TAP) as a kernelized cross-attention module. Theoretical analysis is provided on the attention patch formulation.

All reviewers' major concerns have been addressed. Both the solution and theoretical analysis are solid. The results could be of interest to the general TMLR community. The paper is ready to publish in its current form.